# Social determinants of pertussis and influenza vaccine uptake in pregnancy: a national cohort study in England using electronic health records

Jemma L Walker ![ORCID] ,[1,2,3] Christopher T Rentsch ![ORCID] ,[1,3] Helen I McDonald ![ORCID] ,[1,3] JeongEun Bak ![ORCID] ,[1,3] Caroline Minassian ![ORCID] ,[1] Gayatri Amirthalingam ![ORCID] ,[3,4] Michael Edelstein ![ORCID] ,[1,3,4] Sara Thomas[1,3]

► Prepublication history and supplemental material for this paper is available online. To view these files, please visit the journal online (http://dx.doi. org/10.1136/bmjopen-2020- 046545).

JLW and CTR contributed equally.

[1]Faculty of Epidemiology and Population Health, London School of Hygiene and Tropical Medicine, London, UK
[2]Statistics, Modelling and Economics Department, Public Health England, London, UK
[3]NIHR Health Protection Research Unit in Vaccines and Immunisation, London, UK
[4]Immunisation and Countermeasures Division, National Infection Service, Public Health England, London, UK

**Correspondence to**
Dr Helen I McDonald;
helen.mcdonald@lshtm.ac.uk

## ABSTRACT

**Objective** To examine the social determinants of influenza and pertussis vaccine uptake among pregnant women in England.

**Design** Nationwide population-based cohort study.

**Setting** The study used anonymised primary care data from the Clinical Practice Research Datalink and linked Hospital Episode Statistics secondary care data.

**Participants** Pregnant women eligible for pertussis (2012–2015, n=68 090) or influenza (2010/2011– 2015/2016, n=152 132) vaccination in England.

**Main outcome measures** Influenza and pertussis vaccine uptake.

**Results** Vaccine uptake was 67.3% for pertussis and 39.1% for influenza. Uptake of both vaccines varied by region, with the lowest uptakes in London and the North East. Lower vaccine uptake was associated with greater deprivation: almost 10% lower in the most deprived quintiles compared with the least deprived for influenza (34.5% vs 44.0%), and almost 20% lower for pertussis (57.7% vs 76.0%). Lower uptake for both vaccines was also associated with non-white ethnicity (lowest among women of black ethnicity), maternal age under 20 years and a greater number of children in the household. The associations between all social factors and vaccine uptake were broadly unchanged in fully adjusted models, suggesting the social determinants of uptake were largely independent of one another. Among 3111 women vaccinated against pertussis in their first eligible pregnancy and pregnant again, 1234 (40%) were not vaccinated in their second eligible pregnancy.

**Conclusions** Targeting promotional campaigns to pregnant women who are younger, of non-white ethnicity, with more children, living in areas of greater deprivation or the London or North East regions, has potential to reduce vaccine-preventable disease among infants and pregnant women, and to reduce health inequalities. Vaccination promotion needs to be sustained across successive pregnancies. Further research is needed into whether the effectiveness of vaccine promotion strategies may vary according to social factors.

## INTRODUCTION

Pertussis (whooping cough) and seasonal influenza are vaccine-preventable diseases. Influenza can have severe outcomes among

## Strengths and limitations of this study

► This large cohort study explored the social determinants of influenza and pertussis vaccination among pregnant women across England. It considered a range of social determinants including maternal age, ethnicity, socioeconomic status, number of children in the household and region.

► The Clinical Practice Research Datalink/London School of Hygiene and Tropical Medicine Pregnancy Register was used to ascertain pregnancies and their timing from primary care records using detailed algorithms.

► We were unable to investigate other potential social determinants of uptake not routinely recorded in primary care records such as education or religion.

pregnant women and young infants, including hospitalisation and death.[1] Pertussis can be a serious illness for young infants: a pertussis outbreak in 2012 resulted in 14 infant deaths, most of whom were too young to be vaccinated directly.[2–4] Vaccination in pregnancy reduces influenza-associated hospitalisation among pregnant women,[5] and provides 'passive immunity' to protect infants in the first months of life.[6 7] In England, pertussis vaccination has been offered to women in later stages of pregnancy since 2012 and seasonal influenza vaccination at any stage of pregnancy during influenza season since 2010, with both provided free of charge.[2 8]

Low vaccine uptake during pregnancy is a major public health challenge for high-income countries.[9] According to routine surveillance in 2018/2019, vaccine uptake among pregnant women in England was 68.8% for pertussis and 45.2% for influenza.[10 11] Although comparatively high for a high-income country, this suboptimal uptake still limits the programme's impact and results

in vaccine-preventable deaths among infants of unvaccinated mothers. Studies of determinants of maternal influenza vaccine uptake to date have largely focused on health beliefs.[12] Studies in the USA have found inequalities in vaccine uptake during pregnancy by ethnicity/race, age and insurance status.[13–15] Less is known about the role of social factors in England. During the 2009 influenza pandemic, higher vaccine uptake in pregnancy was associated with higher maternal age, previous deliveries and underlying health conditions but not deprivation.[16] However, ecological studies suggest that both seasonal influenza and pertussis vaccine uptake in pregnancy vary with ethnicity, and are lower in areas with greater deprivation, and are thus sources of health inequalities in infancy.[17 18] Smaller studies of pertussis and seasonal influenza vaccines have suggested deprivation, ethnicity, maternal age and parity or number of children may be factors in maternal vaccine uptake, but have lacked power to describe these associations fully.[19–23] A better understanding of the social determinants of maternal vaccine uptake could inform targeted public health interventions to improve vaccine uptake and reduce health inequalities.

This study aimed to use linked electronic health records to examine the social determinants of influenza and pertussis vaccine uptake among pregnant women in England for the first few years from programme introduction: 2012–2015 for pertussis and 2010/2011–2015/2016 for influenza vaccination.

## METHODS

### Data sources

This historical cohort study used data from the Clinical Practice Research Datalink (CPRD), a quality-assured anonymised primary care patient dataset covering approximately 7% of general practices in England, a representative sample of the population by age and sex.[24 25] Available data include diagnoses and symptoms, prescriptions, immunisations and referrals recorded in primary care. The CPRD/London School of Hygiene and Tropical Medicine (LSHTM) Pregnancy Register details all pregnancies recorded in primary care, identified using detailed algorithms to determine their timing and outcomes.[26] The Pregnancy Register has been found to have a high sensitivity for livebirths (including 90% of all deliveries recorded in secondary care) but may under-record pregnancies which end in a loss.[26 27] For this analysis, we used the Pregnancy Register and CPRD data prelinked to Hospital Episode Statistics (HES) admissions data (for supplementary ethnicity data),[28] and Office of National Statistics (ONS) small-area-level deprivation data.[29]

### Study population

Analysis of pertussis vaccine and seasonal influenza vaccine uptake were conducted separately. For each vaccine, we identified pregnancies eligible for the relevant vaccination among women registered with CPRD, using the Pregnancy Register to identify start and end dates of pregnancies, eligible dates based on gestation, and pregnancy outcomes. Eligible women were registered at one of the 75% of CPRD practices in England which participate in the CPRD data-linkage scheme, for availability of linked HES and ONS data.[24] Vaccine eligibility started on or after 1 October 2012 for the pertussis vaccine analyses, and on or after 1 April 2010 for the seasonal influenza vaccine analyses, reflecting the introduction of vaccination programmes.[2 8] For each vaccine, the first eligible pregnancy for each woman during the follow-up period was used to avoid non-independence in the data.

Vaccination guidelines during the study period suggested women should be offered pertussis vaccination in their third trimester of pregnancy (ideally between 28 and 32 weeks, though it could be offered between 28 and 38 weeks' gestation).[2] For the pertussis vaccine analyses, we included women who delivered a live or stillborn child on or after 26 weeks of pregnancy and followed up for vaccination up to 40 weeks' gestation, which allowed for up to 2 weeks imprecision in the Pregnancy Register estimation of the vaccine eligible period and mirrored the national surveillance approach. The study period ended before the April 2016 change in guidelines recommending vaccination at 16–32 weeks of pregnancy (though it may be given up to delivery), and changes in the commissioning arrangements leading to increased delivery through maternity services from 2016.[2]

Influenza vaccination is recommended at any stage in pregnancy that overlaps with the influenza season.[8] For the influenza vaccine analyses, all pregnancies for which the Pregnancy Register included a known outcome (such as stillbirth, live birth, miscarriage or termination) were included, irrespective of duration of pregnancy, providing the pregnancy overlapped by at least 1 day with the influenza season (1 September to 31 January of each year).

We limited primary analyses for both maternal vaccines to women who registered as patients at the primary care practice by the end of their first trimester, to reduce misclassification of vaccination status. We conducted sensitivity analyses around the study inclusion criteria, which are described later.

### Follow-up period

The study period ranged from 1 October 2012 to 30 September 2015 for pertussis vaccine and 1 September 2010 to 31 January 2016 for influenza vaccine. Start of follow-up was considered the latest date of: start of the study period, practice meeting CPRD quality standards, patient registration at the practice, 11th birthday (dates of birth based on the mid-point of year of birth), 26 weeks' gestation of pregnancy (for pertussis), the start of pregnancy plus 2 weeks (for influenza) or 1 September of each year (for influenza). End of follow-up was the earliest date of: last data collection from the practice, end of linkage to HES, patient transfer out of the practice, 49th birthday, death, receipt of the vaccine of interest, the 40th week of pregnancy (for pertussis), end of pregnancy

(for influenza), end of the study period or 31 January of each year (for influenza).

## Vaccine uptake

Vaccination status for both maternal pertussis and influenza vaccines was extracted from CPRD. For the primary analysis of pertussis vaccine uptake, women were considered vaccinated if they received the vaccine between 26 and 40 weeks of pregnancy gestation, which is similar to the national vaccination guidelines of 28–38 weeks but allows for up to 2 weeks discrepancy in the Pregnancy Register estimation of gestation. Women who were not vaccinated between 26 and 40 weeks of gestation were considered unvaccinated, irrespective of vaccination before 26 weeks or after 40 weeks of gestation. For the primary analysis of influenza vaccine uptake, women were considered vaccinated if they received the vaccine on any day between 1 September and 31 January during their follow-up period. Women with a pregnancy that spanned two influenza seasons (n=19963, 14%) were counted in the denominator of the latter season and considered vaccinated if vaccinated in either season.

## Social characteristics and clinical conditions

We defined social determinants using previously published detailed algorithms.[30] Index of multiple deprivation (IMD, a composite measure of relative deprivation) was assigned in quintiles (quintile one representing least deprived, quintile five most deprived) based on the Lower Super Output Area of the patient's residential address using ONS data.[29] Ethnicity (white, South Asian, black, mixed, other) was defined using primary care records supplemented with linked HES data.[28] Other social factors of interest were defined using CPRD primary care data and comprised: region of residence (London, North East, North West, Yorkshire and The Humber, East Midlands, West Midlands, East of England, South West, South Central and South East Coast), maternal age (based on midpoint of year of birth) and number of children in the household.

For influenza vaccine uptake analyses, whether the individual was in a clinical risk group indicated to receive influenza vaccine was defined according to national guidance,[8] and comprised the following conditions: chronic renal disease, chronic heart disease, chronic respiratory disease, chronic liver disease, diabetes, immunosuppression, chronic neurological disease, asplenia and morbid obesity. Clinical risk groups were identified using Read codes, primary care prescription records (for immunosuppression and asthma), and height and weight records. Body mass index (BMI) was defined using height and weight records using validated methods,[31] and defined based on the record closest to the beginning of pregnancy, allowing measures during the first trimester of pregnancy. Asthma was defined as an asthma diagnosis and either any history of an emergency hospital admission for asthma, or any inhaled or oral steroid prescription in the previous 12 months. The algorithms used for immunosuppression are described in previous studies[32]; codelists for other conditions are available online (https://doiorg/1017037/DATA00001907).

## Statistical analysis

Parallel analyses were conducted for pertussis and influenza vaccine uptake. For each vaccine, a complete case analysis (excluding women with no ethnicity recorded in the main analysis) using multivariable logistic regression was used to estimate associations between vaccine uptake and social determinants. Our modelling strategy followed a previously adapted version[33] of a conceptual framework to analyse the hierarchical inter-relationships between distal and proximate social determinants with vaccine uptake (online supplemental file 1).[34] We first fitted a 'minimally adjusted' model to estimate associations between each social determinant and vaccine uptake adjusted for year (calendar year for pertussis, financial year for influenza to reflect the influenza season) to adjust for secular trends as an a priori confounder. We then fitted five further sequential models. Models 1–3 explored the social determinants of uptake from distal to proximal. Model 4 and the BMI model explored the extent to which these were mediated by clinical conditions (for influenza), and mediated and/or confounded by BMI (for both vaccines).

In model 1 we assessed associations between vaccine uptake and the distal determinants IMD, region and ethnicity, mutually adjusted and adjusted for year. In model 2 the intermediate variable maternal age was added alongside the variables in model 1 to determine to what extent this explained any effect of the distal variables. Model 3 comprised the variables in model 2 and the proximate variable number of children, to investigate whether this mediated the effect of the distal and intermediate variables. For influenza uptake modelling, we further added clinical risk group as a potential mediator of the social characteristics (model 4). Finally, we repeated complete case analyses additionally excluding women with no recorded BMI for all four models, adding a further model (BMI model) that additionally adjusted for BMI, which may both mediate and confound the effect of social characteristics and clinical conditions.

All analyses were conducted using Stata V.15 (StataCorp).

## Missing data and sensitivity analyses

Primary analyses were conducted on women who had nonmissing ethnicity and who were registered with an up-to-standard CPRD practice by the end of their first trimester. Other than ethnicity, only BMI had missing data.

We performed descriptive and sensitivity analyses to understand how estimates of vaccine uptake and associations with social determinants might be affected by missing data or study inclusion criteria. First, we examined the distribution of social determinants among women with and without recorded ethnicity. Second, we compared estimates from minimally and fully adjusted

models from the primary analyses with sensitivity analyses including women who registered with an up-to-standard practice by the end of pregnancy (instead of end of first trimester) for both vaccines. For the pertussis analyses, we further ran minimally and fully adjusted models that mirrored national surveillance criteria of immunisation at 28–38 weeks' gestation, to assess the impact of allowing a 2-week window for imprecise estimation of gestation in our primary analysis. For the influenza analyses, we further ran models that included pregnancies with no recorded outcome, as well as models that extended the influenza season through 31 March of each year. Finally, for both pertussis and influenza analyses, we fitted random effects models to test for clustering by general practice.

### Secondary analysis of sequential pregnancies

In response to the finding that vaccine uptake declined with greater number of children in the household, a post-hoc secondary analysis was added investigating the social determinants associated with vaccination in a second eligible pregnancy among women who had received pertussis vaccination in their first eligible pregnancy. This analysis focused on pertussis vaccination, as influenza vaccination uptake may depend on the extent and timing of the overlap of pregnancy with the influenza season, severity of the influenza season and timing of vaccine availability, reducing the number of eligible sequential pregnancies and increasing the complexity of external factors which may affect a women's vaccine uptake across sequential pregnancies. Logistic regression with likelihood ratio tests were used to model and test minimally adjusted and fully adjusted (model 3) associations between the outcome (vaccination in the second eligible pregnancy) and social determinants measured at baseline of the first eligible pregnancy, as well as additionally adjusting for the time interval between the end of the first pregnancy and the start of the next.

### RESULTS
### Sample characteristics

A total of 68 090 women from 402 general practices were eligible for the pertussis vaccine analysis, and 152 132 women from 456 general practices were eligible for the influenza vaccine analysis during the study period (2012–2015 for pertussis and 2010/2011–2015/2016 for influenza). Many women were eligible to be offered both pertussis and influenza vaccinations during the study: 66 143 women were included in both analytic samples (97.1% of the pertussis vaccine cohort and 43.5% of the influenza cohort). There were 5553 (8.9%) and 11 991 (7.9%) women from the pertussis and influenza vaccine analyses, respectively, who had missing ethnicity and were excluded from analysis.

Compared with women with recorded ethnicity, women with missing ethnicity were more likely to have an eligible pregnancy later in the study period, reside in South Central or South East Coast regions of England,

have no children living in their household and to have missing BMI information. Vaccine uptake was similar between women with recorded versus missing ethnicity for pertussis (67.3% vs 68.2) and influenza (39.1% vs 40.4%) (all p<0.001, online supplemental file 1).

### Primary analyses: pertussis vaccination

Among 62 537 eligible women with recorded ethnicity, maternal pertussis vaccine uptake increased each year, reaching 71.7% in 2015 (table 1). Uptake was also highest in the least deprived areas (76.0%, figure 1) and East and West Midlands (74.5% and 72.9%, respectively), and among women of white ethnicity (69.0%), aged 30–35 years (70.8%), who had no other children living in household (74.4%), who were of normal weight or overweight (69.2% and 69.3%, respectively).

After adjusting for calendar year, those who resided in the most deprived areas had less than half the odds of vaccine uptake compared with those in the least deprived areas, and those in all regions of England apart from the North East had increased odds of uptake compared with London (table 1). Pertussis vaccination uptake was appreciably lower among all non-white ethnic groups, with reduced odds of between 24% (South Asian) and 55% (black ethnicity) compared with those of white ethnicity. The odds of vaccination increased non-linearly with maternal age; compared with women aged 20–24 years, women who were <20 years had 21% lower odds of receiving vaccination and there was an increased likelihood of vaccination among women aged ≥25 years, reaching 54% increased odds of uptake among those aged 30–35 years. Uptake decreased linearly with increasing numbers of children living in the household; 33% less likely among women with one child, 53% less likely among women with two children and 65% less likely among women with three or more children (table 1). Among the 55 871 women with available BMI data, calendar-year adjusted uptake was 29% less likely among women whose BMI was classified as underweight and 18% less likely among women classified as obese, compared with women with normal BMI (table 1).

Associations in the minimally adjusted models were largely unchanged after additionally adjusting for IMD, region and ethnicity (model 1), maternal age (model 2) and number of children (model 3). Associations were slightly attenuated (>10% change) for some regions in England (ie, East of England, South Central and South East Coast) in model 1 and model 2, but not in model 3. Similarly, associations of pertussis uptake were marginally attenuated in non-white ethnic groups by adjustment for IMD and region (model 2). However, strong evidence of all these associations remained. Model estimates were also robust to the additional adjustment for BMI in the subset of women with non-missing BMI (online supplemental file 1).

**Table 1** Pertussis vaccine uptake by social characteristics among pregnant women in England, 2012–2015

| | Total (column %) | Received pertussis vaccine unadjusted coverage (row %) | Minimally adjusted for year 'minimally adjusted' OR (95% CI) | Model 1 additionally adjusted for IMD, region and ethnicity OR (95% CI) | Model 2 additionally adjusted for maternal age OR (95% CI) | Model 3 additionally adjusted for number of children 'fully adjusted' OR (95% CI) |
|---|---|---|---|---|---|---|
| **Year** | | | | | | |
| 2012 | 6717 (10.7) | 3809 (56.7) | 1 | 1 | 1 | 1 |
| 2013 | 24 657 (39.4) | 16 749 (67.9) | 1.62 (1.53 to 1.71) | 1.66 (1.57 to 1.75) | 1.66 (1.57 to 1.75) | 1.69 (1.60 to 1.79) |
| 2014 | 20 148 (32.2) | 13 638 (67.7) | 1.60 (1.51 to 1.69) | 1.63 (1.54 to 1.73) | 1.63 (1.54 to 1.73) | 1.66 (1.57 to 1.76) |
| 2015 | 11 015 (17.6) | 7903 (71.7) | 1.94 (1.82 to 2.07) | 2.00 (1.87 to 2.13) | 2.00 (1.87 to 2.13) | 2.03 (1.90 to 2.17) |
| **Index of multiple deprivation (IMD) quintile** | | | | | | |
| Least deprived | 13 285 (21.2) | 10 090 (76.0) | 1 | 1 | 1 | 1 |
| 2 | 11 335 (18.1) | 8064 (71.1) | 0.78 (0.74 to 0.83) | 0.79 (0.74 to 0.83) | 0.80 (0.75 to 0.85) | 0.81 (0.76 to 0.86) |
| 3 | 12 933 (20.7) | 8807 (68.1) | 0.68 (0.64 to 0.71) | 0.68 (0.64 to 0.72) | 0.70 (0.66 to 0.74) | 0.73 (0.69 to 0.77) |
| 4 | 12 973 (20.7) | 8205 (63.2) | 0.54 (0.52 to 0.57) | 0.56 (0.53 to 0.59) | 0.59 (0.56 to 0.62) | 0.64 (0.60 to 0.67) |
| Most deprived | 12 011 (19.2) | 6933 (57.7) | 0.43 (0.41 to 0.46) | 0.45 (0.42 to 0.47) | 0.48 (0.45 to 0.51) | 0.54 (0.51 to 0.57) |
| **Region** | | | | | | |
| London | 11 894 (19.0) | 7239 (60.9) | 1 | 1 | 1 | 1 |
| North East | 1185 (1.9) | 687 (58.0) | 0.91 (0.81 to 1.03) | 0.96 (0.85 to 1.09) | 1.00 (0.88 to 1.13) | 1.04 (0.92 to 1.19) |
| North West | 8835 (14.1) | 5873 (66.5) | 1.29 (1.22 to 1.36) | 1.28 (1.20 to 1.35) | 1.30 (1.22 to 1.38) | 1.36 (1.27 to 1.44) |
| Yorkshire and The Humber | 1000 (1.6) | 699 (69.9) | 1.51 (1.31 to 1.74) | 1.46 (1.27 to 1.69) | 1.51 (1.30 to 1.74) | 1.54 (1.33 to 1.79) |
| East Midlands | 326 (0.5) | 243 (74.5) | 2.18 (1.69 to 2.81) | 2.24 (1.73 to 2.90) | 2.30 (1.78 to 2.98) | 2.38 (1.84 to 3.09) |
| West Midlands | 7050 (11.3) | 5046 (71.6) | 1.64 (1.54 to 1.75) | 1.58 (1.48 to 1.69) | 1.62 (1.52 to 1.73) | 1.72 (1.61 to 1.84) |
| East of England | 5568 (8.9) | 4058 (72.9) | 1.75 (1.63 to 1.88) | 1.50 (1.40 to 1.61) | 1.52 (1.41 to 1.63) | 1.57 (1.46 to 1.69) |
| South West | 7002 (11.2) | 4800 (68.6) | 1.43 (1.34 to 1.52) | 1.32 (1.24 to 1.41) | 1.35 (1.26 to 1.44) | 1.43 (1.33 to 1.52) |
| South Central | 10 381 (16.6) | 7185 (69.2) | 1.45 (1.37 to 1.53) | 1.19 (1.12 to 1.26) | 1.21 (1.15 to 1.29) | 1.28 (1.21 to 1.36) |
| South East Coast | 9296 (14.9) | 6269 (67.4) | 1.33 (1.26 to 1.41) | 1.10 (1.04 to 1.17) | 1.12 (1.06 to 1.19) | 1.19 (1.12 to 1.26) |
| **Ethnicity** | | | | | | |
| White | 52 598 (84.1) | 36 272 (69.0) | 1 | 1 | 1 | 1 |
| South Asian | 4692 (7.5) | 2951 (62.9) | 0.76 (0.71 to 0.81) | 0.83 (0.78 to 0.88) | 0.79 (0.74 to 0.85) | 0.83 (0.78 to 0.88) |
| Black | 2583 (4.1) | 1294 (50.1) | 0.45 (0.41 to 0.48) | 0.58 (0.54 to 0.64) | 0.56 (0.52 to 0.61) | 0.61 (0.56 to 0.67) |
| Mixed | 922 (1.5) | 549 (59.5) | 0.65 (0.57 to 0.74) | 0.72 (0.63 to 0.82) | 0.71 (0.62 to 0.82) | 0.72 (0.63 to 0.83) |
| Other | 1742 (2.8) | 1033 (59.3) | 0.65 (0.59 to 0.72) | 0.73 (0.66 to 0.80) | 0.70 (0.63 to 0.77) | 0.68 (0.62 to 0.75) |
| **Maternal age, years** | | | | | | |
| <20 | 2079 (3.3) | 1153 (55.5) | 0.79 (0.72 to 0.87) | | 0.80 (0.73 to 0.89) | 0.81 (0.73 to 0.89) |
| 20–24 | 8848 (14.1) | 5416 (61.2) | 1 | | 1 | 1 |
| 25–29 | 16 696 (26.7) | 11 166 (66.9) | 1.27 (1.21 to 1.34) | | 1.24 (1.18 to 1.31) | 1.29 (1.22 to 1.36) |
| 30–35 | 20 294 (32.5) | 14 376 (70.8) | 1.54 (1.46 to 1.62) | | 1.43 (1.35 to 1.51) | 1.55 (1.47 to 1.64) |
| ≥35 | 14 620 (23.4) | 9988 (68.3) | 1.36 (1.29 to 1.44) | | 1.25 (1.18 to 1.32) | 1.42 (1.34 to 1.51) |
| **Number of children** | | | | | | |
| 0 | 26 622 (42.6) | 19 814 (74.4) | 1 | | | 1 |
| 1 | 22 132 (35.4) | 14 673 (66.3) | 0.67 (0.65 to 0.70) | | | 0.65 (0.63 to 0.68) |
| 2 | 8645 (13.8) | 5009 (57.9) | 0.47 (0.45 to 0.49) | | | 0.47 (0.45 to 0.50) |
| ≥3 | 5138 (8.2) | 2603 (50.7) | 0.35 (0.33 to 0.37) | | | 0.37 (0.35 to 0.40) |
| **Body mass index (BMI)** | | | | | | |
| <18.5 underweight | 2063 (3.3) | 1265 (61.3) | 0.71 (0.64 to 0.77) | | | |
| 18.5–24.9 | 29 045 (46.4) | 20 095 (69.2) | 1 | | | |
| 25.0–29.9 overweight | 14 211 (22.7) | 9852 (69.3) | 1.01 (0.96 to 1.05) | | | |

Continued

**Table 1** Continued

| | Total (column %) | Received pertussis vaccine unadjusted coverage (row %) | Minimally adjusted for year 'minimally adjusted' OR (95% CI) | Model 1 additionally adjusted for IMD, region and ethnicity OR (95% CI) | Model 2 additionally adjusted for maternal age OR (95% CI) | Model 3 additionally adjusted for number of children 'fully adjusted' OR (95% CI) |
|---|---|---|---|---|---|---|
| ≥30 obese | 10 552 (16.9) | 6833 (64.8) | 0.82 (0.78 to 0.86) | | | |
| Missing | 6666 (10.7) | 4054 (60.8) | | | | |

N=62 537 from 402 practices. Overall vaccine uptake 42 099 (67.3%).
All models include women who registered before the end of the first trimester and delivered a live or stillborn child on or after 26 weeks of pregnancy and exclude those with missing ethnicity; minimally adjusted model of BMI additionally excludes 6666 women with missing BMI.

## Primary analyses: influenza vaccination

Similar to pertussis vaccination, maternal influenza vaccine uptake was highest (46%) by the end of the study period (the 2015/2016 season) among the 140 141 eligible women with recorded ethnicity (table 2). Uptake was also highest in the least deprived areas (44.0%, figure 1), in the South Central and West Midlands regions (42.6% and 42.2%, respectively), and among women of white ethnicity (39.8%), aged 30–35 years (41.0%), who had no children living in household (43.0%) and who were overweight (40.4%). Influenza vaccination uptake was lowest among women of black ethnicity, with 16% reduced odds of uptake compared with those of White ethnicity. Women who were classified as being in a clinical risk group had the highest influenza vaccine uptake (50.9%) out of all subgroups.

Findings of associations between social determinants and influenza vaccine uptake were largely the same as those with pertussis uptake (table 2). Women were 65% more likely to receive the influenza vaccination in the 2015/2016 season compared with the 2010/2011 season. Similarly, in influenza-season adjusted models, women who resided in the most deprived areas had 29% lower odds of receiving vaccination, and women in all regions outside of London were more likely to be vaccinated. Associations with ethnicity, maternal age, number of children and BMI also mirrored those found in the pertussis uptake models, although the lower uptake seen with women of non-white ethnicity was less marked than that

seen for pertussis vaccination. Women identified as being in a clinical risk group for influenza were 69% more likely to be vaccinated than those not in a clinical risk group. Associations were robust throughout all subsequent models except for South Asian ethnicity and South East Coast regional residence, and remained after additional adjustment for clinical risk group in model 4 (table 2). Model estimates were also robust to the additional adjustment for BMI in the model excluding those with missing BMI (online supplemental file 1).

## Sensitivity analyses

Directions of associations and conclusions were robust to all sensitivity analysis for pertussis vaccination (online supplemental file 1) and influenza vaccination (online supplemental file 1), and we found no evidence of clustering at the practice level in the primary analysis models for either pertussis or influenza uptake (p=0.07, 95% CI 0.06 to 0.09 for pertussis, p=0.03, 95% CI 0.03 to 0.03 for influenza).

## Secondary analysis

Among women who were included in the main study, there were 3111 women who received pertussis vaccination in their first eligible pregnancy and who completed a second eligible pregnancy within the study period. Among these, 1234 (39.7%) were not vaccinated in their second eligible pregnancy. Social determinants of vaccine uptake among women who had previously received vaccination in pregnancy were similar to those in the main analysis, with lower uptake in the second eligible pregnancy associated with younger maternal age at the first pregnancy, a greater number of children in the household and a longer interval between pregnancies (online supplemental file 1).

## DISCUSSION

Vaccine uptake in pregnancy over the study period was 67.3% for pertussis and 39.1% for influenza. Lower vaccine uptake was associated with greater deprivation: the gap in uptake between the least and most deprived quintiles was almost 10% for influenza, and almost 20% for pertussis. Lower uptake was also associated with non-white ethnicity (particularly black ethnicity), maternal age under 20 years and greater number of children in

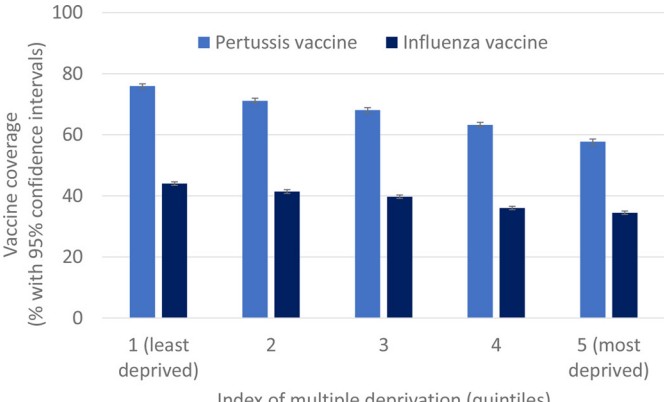

**Figure 1** Unadjusted pertussis and influenza vaccine coverage in pregnancy, by deprivation.

**Table 2** Influenza vaccine uptake by social characteristics among pregnant women in England, 2010/2011–2015/2016

| | Total (column %) | Received influenza vaccine unadjusted coverage (row %) | Minimally adjusted for year 'minimally adjusted' OR (95% CI) | Model 1 additionally adjusted for IMD, region and ethnicity OR (95% CI) | Model 2 additionally adjusted for maternal age OR (95% CI) | Model 3 additionally adjusted for number of children OR (95% CI) | Model 4 additionally adjusted for clinical risk group 'fully adjusted' OR (95% CI) |
|---|---|---|---|---|---|---|---|
| Season | | | | | | | |
| 2010 | 34373 (24.5) | 11703 (34.0) | 1 | 1 | 1 | 1 | 1 |
| 2011 | 32258 (23.0) | 10151 (31.5) | 0.89 (0.86 to 0.92) | 0.89 (0.86 to 0.92) | 0.89 (0.86 to 0.92) | 0.89 (0.86 to 0.92) | 0.89 (0.86 to 0.92) |
| 2012 | 26750 (19.1) | 12236 (45.7) | 1.63 (1.58 to 1.69) | 1.66 (1.61 to 1.72) | 1.66 (1.61 to 1.72) | 1.64 (1.59 to 1.70) | 1.65 (1.60 to 1.71) |
| 2013 | 21029 (15.0) | 8815 (41.9) | 1.40 (1.35 to 1.45) | 1.43 (1.38 to 1.48) | 1.42 (1.37 to 1.47) | 1.39 (1.35 to 1.45) | 1.40 (1.35 to 1.45) |
| 2014 | 15712 (11.2) | 7319 (46.6) | 1.69 (1.63 to 1.76) | 1.74 (1.67 to 1.80) | 1.73 (1.67 to 1.80) | 1.69 (1.63 to 1.76) | 1.70 (1.63 to 1.76) |
| 2015 | 10019 (7.1) | 4613 (46.0) | 1.65 (1.58 to 1.73) | 1.72 (1.65 to 1.80) | 1.72 (1.64 to 1.80) | 1.68 (1.60 to 1.76) | 1.68 (1.60 to 1.76) |
| Index of multiple deprivation (IMD) quintile | | | | | | | |
| Least deprived | 28956 (20.7) | 12744 (44.0) | 1 | 1 | 1 | 1 | 1 |
| 2 | 25424 (18.1) | 10533 (41.4) | 0.90 (0.87 to 0.93) | 0.91 (0.88 to 0.94) | 0.92 (0.89 to 0.95) | 0.93 (0.89 to 0.96) | 0.92 (0.89 to 0.95) |
| 3 | 29368 (21.0) | 11670 (39.7) | 0.84 (0.81 to 0.86) | 0.84 (0.82 to 0.87) | 0.86 (0.83 to 0.89) | 0.88 (0.85 to 0.91) | 0.88 (0.85 to 0.91) |
| 4 | 28520 (20.4) | 10278 (36.0) | 0.71 (0.69 to 0.74) | 0.72 (0.69 to 0.74) | 0.74 (0.71 to 0.77) | 0.77 (0.74 to 0.79) | 0.76 (0.74 to 0.79) |
| Most deprived | 27873 (19.9) | 9612 (34.5) | 0.67 (0.65 to 0.70) | 0.66 (0.64 to 0.68) | 0.69 (0.66 to 0.71) | 0.73 (0.70 to 0.76) | 0.72 (0.70 to 0.75) |
| Region | | | | | | | |
| London | 26171 (18.7) | 9146 (34.9) | 1 | 1 | 1 | 1 | 1 |
| North East | 2758 (2.0) | 989 (35.9) | 1.11 (1.02 to 1.21) | 1.16 (1.07 to 1.27) | 1.19 (1.09 to 1.29) | 1.21 (1.11 to 1.31) | 1.21 (1.11 to 1.31) |
| North West | 19060 (13.6) | 7870 (41.3) | 1.37 (1.32 to 1.42) | 1.39 (1.33 to 1.45) | 1.40 (1.35 to 1.46) | 1.43 (1.37 to 1.49) | 1.42 (1.36 to 1.47) |
| Yorkshire and The Humber | 2840 (2.0) | 1090 (38.4) | 1.27 (1.18 to 1.38) | 1.24 (1.15 to 1.35) | 1.26 (1.16 to 1.37) | 1.26 (1.16 to 1.37) | 1.26 (1.16 to 1.37) |
| East Midlands | 1940 (1.4) | 717 (37.0) | 1.33 (1.21 to 1.47) | 1.37 (1.24 to 1.51) | 1.39 (1.26 to 1.53) | 1.41 (1.27 to 1.55) | 1.40 (1.27 to 1.55) |
| West Midlands | 15846 (11.3) | 6692 (42.2) | 1.41 (1.35 to 1.46) | 1.40 (1.34 to 1.46) | 1.41 (1.36 to 1.47) | 1.44 (1.38 to 1.51) | 1.43 (1.37 to 1.49) |
| East of England | 13695 (9.8) | 5468 (39.9) | 1.31 (1.26 to 1.37) | 1.23 (1.18 to 1.29) | 1.24 (1.19 to 1.29) | 1.25 (1.20 to 1.31) | 1.24 (1.19 to 1.30) |
| South West | 16546 (11.8) | 6504 (39.3) | 1.25 (1.20 to 1.31) | 1.22 (1.17 to 1.28) | 1.24 (1.19 to 1.29) | 1.27 (1.21 to 1.32) | 1.25 (1.20 to 1.31) |
| South Central | 21435 (15.3) | 9125 (42.6) | 1.42 (1.36 to 1.47) | 1.30 (1.25 to 1.35) | 1.31 (1.26 to 1.36) | 1.34 (1.29 to 1.39) | 1.33 (1.28 to 1.38) |
| South East Coast | 19850 (14.2) | 7236 (36.5) | 1.06 (1.02 to 1.10) | 0.99 (0.95 to 1.03) | 1.00 (0.96 to 1.04) | 1.02 (0.98 to 1.06) | 1.02 (0.98 to 1.06) |
| Ethnicity | | | | | | | |
| White | 117469 (83.8) | 46781 (39.8) | 1 | 1 | 1 | 1 | 1 |
| South Asian | 10827 (7.7) | 4103 (37.9) | 0.92 (0.88 to 0.95) | 0.98 (0.94 to 1.02) | 0.96 (0.92 to 1.00) | 0.98 (0.94 to 1.02) | 0.99 (0.95 to 1.03) |
| Black | 5853 (4.2) | 1837 (31.4) | 0.67 (0.64 to 0.71) | 0.81 (0.76 to 0.86) | 0.80 (0.75 to 0.85) | 0.83 (0.78 to 0.88) | 0.83 (0.78 to 0.88) |
| Mixed | 2094 (1.5) | 757 (36.2) | 0.84 (0.77 to 0.92) | 0.90 (0.82 to 0.99) | 0.90 (0.82 to 0.99) | 0.91 (0.83 to 0.99) | 0.91 (0.83 to 0.99) |
| Other | 3898 (2.8) | 1359 (34.9) | 0.79 (0.73 to 0.84) | 0.85 (0.80 to 0.91) | 0.84 (0.78 to 0.90) | 0.83 (0.78 to 0.89) | 0.85 (0.79 to 0.91) |
| Maternal age, years | | | | | | | |
| <20 | 5536 (4.0) | 1817 (32.8) | 0.87 (0.81 to 0.92) | 0.87 (0.82 to 0.93) | 0.87 (0.82 to 0.93) | 0.87 (0.82 to 0.93) | 0.87 (0.82 to 0.93) |

Continued

**Table 2**  Continued

| | Total (column %) | Received influenza vaccine unadjusted coverage (row %) | Minimally adjusted for year 'minimally adjusted' OR (95% CI) | Model 1 additionally adjusted for IMD, region and ethnicity OR (95% CI) | Model 2 additionally adjusted for maternal age OR (95% CI) | Model 3 additionally adjusted for number of children OR (95% CI) | Model 4 additionally adjusted for clinical risk group 'fully adjusted' OR (95% CI) |
|---|---|---|---|---|---|---|---|
| 20–24 | 21663 (15.5) | 7797 (36.0) | 1 | | 1 | 1 | 1 |
| 25–29 | 37985 (27.1) | 14827 (39.0) | 1.13 (1.09 to 1.17) | | 1.11 (1.07 to 1.15) | 1.12 (1.09 to 1.16) | 1.12 (1.08 to 1.16) |
| 30–35 | 43777 (31.2) | 17950 (41.0) | 1.22 (1.18 to 1.26) | | 1.18 (1.14 to 1.22) | 1.21 (1.17 to 1.26) | 1.21 (1.17 to 1.25) |
| ≥35 | 31180 (22.2) | 12446 (39.9) | 1.17 (1.12 to 1.21) | | 1.12 (1.08 to 1.16) | 1.19 (1.15 to 1.24) | 1.18 (1.13 to 1.22) |
| Number of children | | | | | | | |
| 0 | 66112 (47.2) | 28457 (43.0) | 1 | | | 1 | 1 |
| 1 | 45969 (32.8) | 17092 (37.2) | 0.80 (0.78 to 0.82) | | | 0.80 (0.78 to 0.82) | 0.80 (0.78 to 0.82) |
| 2 | 18192 (13.0) | 6242 (34.3) | 0.71 (0.68 to 0.73) | | | 0.72 (0.69 to 0.74) | 0.71 (0.69 to 0.74) |
| ≥3 | 9868 (7.0) | 3046 (30.9) | 0.61 (0.58 to 0.63) | | | 0.63 (0.60 to 0.66) | 0.62 (0.59 to 0.65) |
| Clinical risk group recommended for influenza vaccination | | | | | | | |
| No | 130160 (92.9) | 49752 (38.2) | 1 | | | | 1 |
| Yes | 9981 (7.1) | 5085 (50.9) | 1.69 (1.62 to 1.76) | | | | 1.70 (1.63 to 1.77) |
| Body mass index (BMI) | | | | | | | |
| <18.5 underweight | 4865 (3.5) | 1744 (35.8) | 0.85 (0.80 to 0.90) | | | | |
| 18.5–24.9 | 66405 (47.4) | 26331 (39.7) | 1 | | | | |
| 25.0–29.9 overweight | 31855 (22.7) | 12882 (40.4) | 1.04 (1.01 to 1.07) | | | | |
| ≥30 obese | 23142 (16.5) | 9222 (39.8) | 1.00 (0.97 to 1.03) | | | | |
| Missing | 13874 (9.9) | 4658 (33.6) | | | | | |

N=140141 from 456 practices. Overall vaccine uptake 54837 (39.1%).
All models include women who registered before the end of the first trimester, and exclude those with no recorded pregnancy outcome or missing ethnicity; minimally adjusted model of BMI additionally excludes 13874 women with missing BMI.

the household. The associations between all social factors and vaccine uptake were largely independent of one another. Among women eligible for pertussis vaccination in two pregnancies and vaccinated in the first, 40% were not vaccinated in their second eligible pregnancy.

To our knowledge, this is the first large study of fully individual-level social determinants of maternal vaccine uptake of seasonal influenza and pertussis in England. Our findings differ from a large national study which found no association between deprivation and pandemic influenza vaccine uptake in pregnancy (although vaccine uptake did increase with maternal age) but the previous study was in the context of the 2010 influenza pandemic.[16] Both the overall uptakes and the patterns of regional variation are consistent with national surveillance and ecological studies. Lower vaccine uptake in London is seen more widely across the vaccination programme.[10 11 17 18] For influenza vaccine, the denominator may be seen as overinclusive as some women may have only a short time period eligible for vaccination (due to pregnancy loss or limited overlap of pregnancy with influenza season), resulting in a low estimate of uptake. For seasonal influenza and pertussis vaccines, previous studies have generally suggested associations consistent with those we observed for deprivation, ethnicity, maternal age and parity or number of children, but studies have been ecological or pseudo-individualised, or were underpowered for precise estimates.[17–21 23] Our findings in a large and nationally representative dataset demonstrate that each of these factors is an independent individual-level determinant of maternal vaccine uptake, outside of a pandemic context.

The novel finding that 40% of women who had been vaccinated in their first eligible pregnancy were not in their second is surprising, and suggests that low vaccine uptake in pregnancy is not fully determined by fixed maternal attitudes to vaccination, but may reflect healthcare access or awareness of the need for vaccination in each pregnancy.

Strengths of this study include the use of the CPRD/LSHTM Pregnancy Register with linked hospital and mortality data and detailed algorithms to identify pregnancy timings and a range of individual-level social determinants among a nationally representative population.[30]

Key limitations include low representation from some regions (in particular the East Midlands), and that not all potentially relevant social factors were available, such as education and religion. We may have over-estimated vaccine uptake as the pregnancy register may not include all pregnancies which ended in a loss without coming to the attention of healthcare workers. We included only timely pertussis vaccinations (before 40 weeks' gestation) which may result in lower uptake estimates than pertussis vaccine uptake by delivery. Our study was also limited to vaccination recorded in primary care records, which could have resulted in some under-recording of influenza vaccination, although maternity-led vaccination services were rare before 2016, and general practitioners

are required to document vaccinations given outside the surgery. To minimise misclassification, we ended our study period prior to the introduction of pertussis vaccination in antenatal settings.

The large differences we observed in vaccine uptake by deprivation and ethnicity indicate a key opportunity to reduce health inequalities. Targeting interventions and improving access to vaccines through primary care and maternity services for pregnant women who live in more deprived areas, are of non-white ethnicity, younger, or have more children may reduce health inequalities, improve overall vaccine uptake and reduce vaccine-preventable deaths among women and children. In addition to targeted vaccination promotion, wider action is needed to address inequalities in access to timely antenatal care.[35] The drop-off in uptake in second pregnancies suggests a need for awareness-raising of the rationale for passive immunisation of infants and the need for vaccination in each pregnancy. Communications to emphasise the need for vaccination in every pregnancy should be available in a range of locally appropriate languages. Since 2016, pertussis vaccination has been available in maternity services, aiming to increase opportunities for vaccine uptake, and it will be important to ensure that healthcare worker training also captures the importance of vaccination in every pregnancy and to monitor the impact of delivery in alternative settings on inequalities in uptake.

Our study adds to international evidence of health inequalities in vaccination uptake in high-income countries. Studies in the USA have found inequalities in vaccine uptake by insurance type, race/ethnicity and education.[13–15] Our finding of large inequalities in vaccine uptake during pregnancy in England, despite universal healthcare which is free at the point of access, highlights the need for other high-income countries to investigate and address inequalities in vaccine uptake during pregnancy.

Further research is needed into interventions to reduce inequalities in vaccine uptake during pregnancy,[36] to ensure that future vaccine promotion of these and any future maternal vaccination programmes succeed in narrowing rather than widening the large and multi-faceted health inequalities in early years.

**Acknowledgements** This work uses data provided by patients and collected by the NHS as part of their care and support and would not have been possible without access to this data. The NIHR recognises and values the role of patient data, securely accessed and stored, both in underpinning and leading to improvements in research and care.

**Contributors** JLW and ST conceived the main study, and CTR and HIM conceived the secondary analysis. JLW, CTR, HIM, CM and ST designed the study. JLW performed the data extraction and JLW and CTR performed the statistical analyses. JEB, CTR and HIM designed the secondary analysis, for which JEB and HIM performed the statistical analysis. JLW, CTR, HIM, JEB, CM, GA, ME and ST contributed to the interpretation of results. CTR and HIM drafted the manuscript, which JLW, JEB, CM, GA, ME and ST contributed to, revised critically and approved. HIM is the guarantor. The corresponding author (HIM) attests that all listed authors meet authorship criteria and that no others meeting the criteria have been omitted.

**Funding** The research was funded by the National Institute for Health Research (NIHR) Health Protection Research Unit (HPRU) in Immunisation (Grant number IS-HPU1112-10096) at the London School of Hygiene and Tropical Medicine in partnership with Public Health England (PHE). The views expressed are those of the authors and not necessarily those of the NHS, the NIHR, the Department of Health and Social Care, or PHE.

**Competing interests** All authors have completed the ICMJE uniform disclosure form at www.icmje.org/coi_disclosure.pdf and declare: JLW, CTR, HIM and ST had financial support from the National Institute for Health Research (NIHR) Health Protection Research Unit (HPRU) in Immunisation for the submitted work; Public Health England Immunisation and Countermeasures Division has provided vaccine manufacturers with post-marketing surveillance reports on pneumococcal and meningococcal infection which the companies are required to submit to the UK Licensing Authority in compliance with their Risk Management Strategy, and a cost recovery charge is made for these reports; no other relationships or activities that could appear to have influenced the submitted work.

**Patient consent for publication** Not required.

**Ethics approval** The study was approved by the Independent Scientific Advisory Group of the CPRD (ISAC reference 17_030RA2) and the London School of Hygiene and Tropical Medicine Ethics Committee (LSHTM reference 16265). The study protocol was made available to reviewers

**Provenance and peer review** Not commissioned; externally peer reviewed.

**Data availability statement** Data may be obtained from a third party and are not publicly available. The data used for this study were obtained from the Clinical Practice Research Datalink (CPRD). All data are available via an application to the Independent Scientific Advisory Committee (see https://www.cprd.com/Data-access). Data acquisition is associated with a fee and subject to ethics approval.

**ORCID iDs**
Jemma L Walker http://orcid.org/0000-0003-3728-9509
Christopher T Rentsch http://orcid.org/0000-0002-1408-7907
Helen I McDonald http://orcid.org/0000-0003-0576-2015
JeongEun Bak http://orcid.org/0000-0001-7519-7539
Caroline Minassian http://orcid.org/0000-0001-9406-1928
Gayatri Amirthalingam http://orcid.org/0000-0003-2078-0975
Michael Edelstein http://orcid.org/0000-0002-7323-0806

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
