## [Reviewer comments · BMJ Open]

ARTICLE DETAILS

TITLE (PROVISIONAL)	Social determinants of pertussis and influenza vaccine uptake in pregnancy: a national cohort study in England using electronic health records
AUTHORS	Walker, Jemma; Rentsch, Christopher; McDonald, Helen; Bak, JeongEun; Minassian, Caroline; Amirthalingam, Gayatri; Edelstein, Michael; Thomas, Sara

VERSION 1 – REVIEW

REVIEWER	Howe, Anna University of Canterbury, Health Sciences
REVIEW RETURNED	13-Dec-2020

GENERAL COMMENTS	I would like to commend the authors on a well written paper exploring social determinants of maternal vaccination during pregnancy in the English setting. I just have a few minor comments. Line 14: The authors state that the CPRD covers around 7% of GPs in England, some indication of the demographic make-up of this population compared to the rest of the country would be helpful context, especially for overseas readers. Vaccine uptake section - page 9: I'm unclear about how you classified pregnant women who did not receive their pertussis immunisation during 26-40 weeks of gestation? Were there any? If so, were these women excluded from the study or classified as unvaccinated? I'm also unsure why such a "scheduled" based definition was necessary, perhaps it needs to be clearer that what the authors are actually examining is receipt of timely vaccination? Discussion - pages 20-22: I feel the discussion could be strengthened with a few additional points. The study results need to be compared with other international literature on this topic in order to provide wider context and for comparison with other large cohort studies that have been undertaken in this area. Generally a discussion putting your results into context would be helpful especially for international readers. For example discussion around differences in population demographics or health delivery or policy of the various regions of England would be helpful. Why might pertussis uptake be higher in the East and West Midlands? Why higher odds of uptake in the North East compared to London? Why might some regions have attenuated results in Model 1 and 2 but not 3? Similar discussion for influenza. Also, why might some areas be different for pertussis and influenza?
--

REVIEWER	Thompson, Lindsay
-----------------	-------------------

	University of Florida College of Medicine, Pediatrics
REVIEW RETURNED	23-Dec-2020

GENERAL COMMENTS	Summary: The authors use a representative sample of linked EHR data to see what social factors predict vaccination during pregnancy. Its strengths are its completeness of documented analysis as well as in the subsequent analysis of second deliveries- this is a very surprising problem of under-vaccination. Main concerns: Global:  1. The analysis lacks a 'so what' next step guide with the reader somewhat lost in tables and supplementary materials. While obviously heavily centered on UK information where the NHS covers financial access to these vaccinations, I do think there are principals that would generalize to other resource-rich countries. 2. The large number of supplementary tables is distracting. Please limit or provide rationale for their large number? The focus on the supplementary materials distracts from the major findings of the paper. Introduction:  1. The sentence that starts on line 39 "Studies...." Claims that "less is known about social factors." We question that claim as there are a lot of known social barriers to influenza vaccination, unless they meant in England. 2. The sentence beginning at line 6 of page 5 "Pertussis..." claims that pregnant women have severe outcomes with pertussis although that is not supported by the citations listed as these citations only focus on infant outcomes for pertussis. It may be clearer to separate pertussis and influenza effects.  a. Some articles that might help here and with the global point #1.  i. resource that includes the major barriers to both influenza and Tdap vaccines, although it isn't technically the focus of the article:  1. Influenza & Tdap: Koepke R, Schauer SL, Davis JP. Measuring maternal Tdap and influenza vaccination rates: comparison of two population-based methods. Vaccine 2017;35:2298-2302. ii. 2 resources that separately cover barriers to influenza and Tdap vaccines:  1. Influenza: Ding H, Black CL, Ball S, et al. Influenza Vaccination Coverage Among Pregnant Women - United States, 2016-17 Influenza Season. MMWR Morb Mortal Wkly Rep. 2017;66(38):1016–1022. 2. Tdap: Housey M, Zhang F, Miller C, et al. Vaccination with tetanus, diphtheria, and acellular pertussis vaccine of pregnant women enrolled in Medicaid--Michigan, 2011-2013. MMWR Morb Mortal Wkly Rep. 2014;63(38):839-842. 3. The last sentence of the introduction implies that the data are only from one year- I would spell out what years the programme introduction occurred for each. (3 and 6 years). Methods:  1. For the data sources, with 7% of GP covered and 100% of all pregnancies linked, does this mean that roughly 7% of all pregnancies are captured here? An estimate would help the reader. 2. The small list of social determinants is disappointing (although the distal and proximal modeling is helpful). Could any other data sources helped with food or housing insecurity, for example? 3. The inclusion of gestational ages prior to 28 weeks for Pertussis analyses is concerning since the recommendation for the immunization begins at 28 weeks and many women receive the vaccine after that date. Could the authors note how many patients
---

	landed in these two week extra analyses? It is also concerning because the data are not adjusted for gestational age. Other papers have chosen 30 weeks to conservatively estimate these kinds of analyses. Results:  1. For ease of understanding and comparison of the two separate cohorts, a more classic table 1 would help. While the information is currently in both the labeled tables 1 and 2, a summary of the overall and vaccinated groups for both would be helpful. 2. It would be helpful to have one figure that conveys the major findings more easily to readers. Discussion:  1. For a non-English reader, the significance of these regions is unknown yet is a major focus. If the authors believe that the geographical differences are of significant clinical value, it could be helpful to explain these differences for non-English readers. Could the authors provide somewhere (perhaps the discussion?) why they think the regional variations exist? 2. The majority of the discussion seems to be further explanation of their data instead of any policy recommendation or next steps. Currently, the discussion only includes vague action items of how to improve in future without clear future directions. 3. Also, a more in depth review of the differences between deprivation/poverty and race would help the reader understand how independent these variables are. Minor concerns:  1. Abstract: Line 42-3, the percentages are in the wrong order compared to the descriptions. 2. Abstract: line 52: the word 'substantially' is superfluous and we would recommend deleting. 3. Article summary: the 4th bullet does not make sense. Please clarify? 4. Introduction: truly minor- there is an extra space in line 20. 5. Methods: truly minor- there is an extra space on page 10, in line 41. 6. Results: First line of sample characteristics, line 9, even though it takes more words, I would keep the data for pertussis and influenza together. 7. Results: line 16- I would add the % of women in both cohorts. 8. Results: Line 33 on page 15 beginning with "Vaccine..." should include a p-value for clarity. 9. Discussion: page 21, line 43, suggest should be singular. 10. Supplementary Table 5 headings appear to be off center. 11. In Tables, it would be helpful to clarify that it is 95% CI in parentheses. 12. Also in the tables, bolding the Confidence intervals that do not cross 1.0 would be helpful since there are so many data points to digest. A visual/bar graph of these associations could be helpful.
--	--

VERSION 1 – AUTHOR RESPONSE

Reviewer: 1

Dr. Anna Howe, University of Canterbury, University of Auckland

Comments to the Author:

1. I would like to commend the authors on a well written paper exploring social determinants of maternal vaccination during pregnancy in the English setting. I just have a few minor comments.

Author response: Thank you very much, and we really appreciate your helpful comments which we address point by point below.

2. Line 14: The authors state that the CPRD covers around 7% of GPs in England, some indication of the demographic make-up of this population compared to the rest of the country would be helpful context, especially for overseas readers.

Author response: Thank you for pointing out this omission. We had referenced the data resource profile by Herrett et al (ref 21) and have added to the text that the population of CPRD has been found to be nationally representative by age and sex. We hope that is useful as a brief note, and also highlights the availability of more information in the reference by Herrett et al.

3. Vaccine uptake section - page 9: I'm unclear about how you classified pregnant women who did not receive their pertussis immunisation during 26-40 weeks of gestation? Were there any? If so, were these women excluded from the study or classified as unvaccinated? I'm also unsure why such a "scheduled" based definition was necessary, perhaps it needs to be clearer that what the authors are actually examining is receipt of timely vaccination?

Author response: Thank you for raising the need for clarification on the pertussis vaccination status. During the study period, national guidance was that pertussis vaccination should be given between 28-32 week's gestation, although it could be offered up to 38 weeks. We were therefore aiming to describe receipt of vaccination within the time window eligible under vaccination guidance during the study period, and considered women unvaccinated if they did not receive a vaccine during this time period. Women who received a vaccine only before 26 weeks or after 40 weeks were considered unvaccinated in the analysis.

We have clarified this in the methods:

“For the primary analysis of pertussis vaccine uptake, women were considered vaccinated if they received the vaccine between 26 and 40 weeks of pregnancy gestation, which is similar to the national vaccination guidelines of 28 to 38 weeks but allows for up to two weeks discrepancy in the Pregnancy Register estimation of gestation. **Women who were not vaccinated between 26 and 40 weeks of gestation were considered unvaccinated, irrespective of vaccination before 26 weeks or after 40 weeks of gestation.**”

“Vaccination guidelines during the study period suggested women be offered pertussis vaccination in their third trimester of pregnancy (ideally between 28-32 weeks, **though it could be offered between 28-38 weeks' gestation**).^{2, 8} **For the pertussis vaccine analyses, we included women who delivered a live-or stillborn child on or after 26 weeks of pregnancy and followed up for vaccination up to 40 weeks' gestation, which allowed for up to 2 weeks imprecision in the Pregnancy Register estimation of the vaccine eligible period and mirrored the national surveillance approach.** The study period ended before the April 2016 change in guidelines recommending vaccination at 16-32 weeks of pregnancy (though it may be given up to delivery), and changes in the commissioning arrangements leading to increased delivery through maternity services from 2016.²”

We have added a comment to the discussion on this:

“We included only timely pertussis vaccinations (before 40 weeks' gestation) which may result in lower uptake estimates than pertussis vaccine uptake by delivery.”

4. Discussion - pages 20-22: I feel the discussion could be strengthened with a few additional points. The study results need to be compared with other international literature on this topic in order to provide wider context and for comparison with other large cohort studies that have been undertaken in this area. Generally a discussion putting your results into context would be helpful especially for international readers. For example discussion around differences in population demographics or health delivery or policy of the various regions of England would be helpful. Why might pertussis uptake be higher in the East and West Midlands? Why higher odds of uptake in the North East compared to London? Why might some regions have attenuated results in Model 1 and 2 but not 3? Similar discussion for influenza. Also, why might some areas be different for pertussis and influenza?

Author response: Thank you for these very helpful suggestions to improve the discussion. Both reviewers highlighted the need to strengthen the discussion, and in particular to comment on international relevance. We have edited the discussion throughout to reduce the repetition of results, add to the context of our findings, and include clearer recommendations of priorities for policy and research. From an English perspective, the regional variation we found is mostly of interest as indicating that our findings align with national surveillance (which reports regional variation), and so we have added a brief comment that vaccine uptake is lower in London across the vaccination programme but have dwelled more on the new information on inequalities which our study adds. We hope that the edited discussion is stronger and highlights the key points more clearly.

Reviewer: 2

Dr. Lindsay Thompson, University of Florida College of Medicine

Comments to the Author:

RE: MS# bmjopen-2020-046545

Summary: The authors use a representative sample of linked EHR data to see what social factors predict vaccination during pregnancy. Its strengths are its completeness of documented analysis as well as in the subsequent analysis of second deliveries- this is a very surprising problem of under-vaccination.

Author response: Thank you very much for your helpful comments on the manuscript, which we address point by point below.

Main concerns

Global:

1. The analysis lacks a 'so what' next step guide with the reader somewhat lost in tables and supplementary materials. While obviously heavily centered on UK information where the NHS covers financial access to these vaccinations, I do think there are principals that would generalize to other resource-rich countries.

Author response: Both reviewers highlighted the need to strengthen the discussion, and in particular to comment on international relevance. We have reduced the volume of text discussing the supplementary materials (please see the response to your next point) and edited the discussion throughout to reduce the repetition of results, add to the context of our findings, and include clearer recommendations of priorities for policy and research. In particular we have made use of the papers you kindly pointed us to, in the discussion:

“Our study adds to international evidence of health inequalities in vaccination uptake in high-income countries. Studies in the United States have found inequalities in vaccine uptake by insurance type, race/ethnicity and education.¹³⁻¹⁵ Our finding of large inequalities in vaccine uptake during pregnancy in England, despite universal healthcare which is free at the point of access, highlights the need for other high-income countries to investigate and address inequalities in vaccine uptake during pregnancy.”

We hope that the edited discussion is stronger and highlights the key points more clearly.

2. The large number of supplementary tables is distracting. Please limit or provide rationale for their large number? The focus on the supplementary materials distracts from the major findings of the paper.

Author response: We would like to include the findings from our supplementary tables, to demonstrate that our findings were robust to different assumptions and decisions in the definition of study eligibility and vaccination. However, we agree that they should not distract from the major findings. We have therefore edited the text in the manuscript to provide only the key messages from sensitivity analyses, and it now reads:

“Directions of associations and conclusions were robust to all sensitivity analysis for pertussis vaccination (**Supplementary Table 5**) and influenza vaccination (**Supplementary Table 6**), and we found no evidence of clustering at the practice level in the primary analysis models for either pertussis or influenza uptake ($\rho=0.07$, 95% CI 0.06-0.09 for pertussis, $\rho=0.03$, 95% CI 0.03-0.03 for influenza).”

We hope this helps to focus the manuscript more clearly on the main findings.

Introduction:

1. The sentence that starts on line 39 “Studies....” Claims that “less is known about social factors.”

We question that claim as there are a lot of known social barriers to influenza vaccination, unless they meant in England.

Author response: We had focused on the UK since social determinants of vaccination may be setting-specific, but we agree that it would greatly improve the value of this manuscript to compare to international literature. Thank you for pointing this out, and for so kindly including several relevant references to the US below. We have edited the introduction to read:

“Studies in the United States have found inequalities in vaccine uptake during pregnancy by ethnicity/race, age and insurance status. Less is known about the role of social factors in England.”

We describe edits to the discussion in response to your global point #1 above.

2. The sentence beginning at line 6 of page 5 “Pertussis...” claims that pregnant women have severe outcomes with pertussis although that is not supported by the citations listed as these citations only focus on infant outcomes for pertussis. It may be clearer to separate pertussis and influenza effects.

Author response: Thank you, we had tried to be too brief. We have now expanded this to read:

“Pertussis (whooping cough) and seasonal influenza are vaccine-preventable diseases.

Influenza can have severe outcomes among pregnant women and young infants, including hospitalisation and death.¹ Pertussis can be a serious illness for young infants: a pertussis outbreak in 2012 resulted in 14 infant deaths, most of whom were too young to be vaccinated directly.²⁻⁴”

a. Some articles that might help here and with the global point #1.

i. resource that includes the major barriers to both influenza and Tdap vaccines, although it isn't technically the focus of the article:

1. Influenza & Tdap: Koepke R, Schauer SL, Davis JP. Measuring maternal Tdap and influenza vaccination rates: comparison of two population-based methods. *Vaccine* 2017;35:2298-2302.

ii. 2 resources that separately cover barriers to influenza and Tdap vaccines:

1. Influenza: Ding H, Black CL, Ball S, et al. Influenza Vaccination Coverage Among Pregnant Women - United States, 2016-17 Influenza Season. *MMWR Morb Mortal Wkly Rep.* 2017;66(38):1016–1022.

2. Tdap: Housey M, Zhang F, Miller C, et al. Vaccination with tetanus, diphtheria, and acellular pertussis vaccine of pregnant women enrolled in Medicaid--Michigan, 2011-2013. *MMWR Morb Mortal Wkly Rep.* 2014;63(38):839-842.

Author response: Thank you, we really appreciate the helpful references of studies describing social factors associated with vaccination during pregnancy in the US. We have found them useful in the introduction and discussion, as described above in response to your earlier comments.

3. The last sentence of the introduction implies that the data are only from one year- I would spell out what years the programme introduction occurred for each. (3 and 6 years).

Author response: To improve the clarify of the manuscript, we have edited this to read:

“This study aimed to use linked electronic health records to examine the social determinants of influenza and pertussis vaccine uptake among pregnant women in England for the first few years from programme introduction: 2012 to 2015 for pertussis and 2010/11 to 2015/16 for influenza vaccination.”

Methods:

1. For the data sources, with 7% of GP covered and 100% of all pregnancies linked, does this mean that roughly 7% of all pregnancies are captured here? An estimate would help the reader.

Author response: We have added to the methods a brief summary of the findings of a validation of the register over the original time period of the register from 1987 to 2018 (Minassian et al.) and (since the data have improved since 1987) a paper which used the register to estimate the prevalence of pregnancy in March 2014.

“The Pregnancy Register has a high sensitivity for livebirths but may under-record pregnancies which end in a loss.^{26, 27}”

We have also added a comment to the limitations section of the discussion:

“We may have over-estimated vaccine uptake as the pregnancy register may not include all pregnancies which ended in a loss without coming to the attention of healthcare workers.”

2. The small list of social determinants is disappointing (although the distal and proximal modeling is helpful). Could any other data sources help with food or housing insecurity, for example?

Author response: This analysis builds on previous work by Anu Jain and Sara Thomas to map and quality appraise the information on social determinants which can be obtained using the Clinical Practice Research Datalink (including the currently available linked datasets) (reference 26). Sadly it was not possible within the scope of this project to conduct any new data linkage. We agree that other social factors are likely to be relevant to vaccine uptake, and highlight this as a limitation in the discussion, with religion and education as examples.

3. The inclusion of gestational ages prior to 28 weeks for Pertussis analyses is concerning since the recommendation for the immunization begins at 28 weeks and many women receive the vaccine after that date. Could the authors note how many patients landed in these two week extra analyses? It is also concerning because the data are not adjusted for gestational age. Other papers have chosen 30 weeks to conservatively estimate these kinds of analyses.

Author response: Our rationale for starting follow up at an estimated 26 weeks' gestation was to allow up to two weeks' imprecision in the estimate of the gestational age using the Pregnancy Register. As a sensitivity analysis we further ran minimally and fully adjusted models that mirrored national surveillance criteria of immunisation at 28-38 weeks' gestation, to assess the impact of allowing that two-week window for imprecise estimation of gestation in our primary analysis. The findings (presented in Supplementary Table 5) were unchanged, and we hope this is reassuring that our decision to follow up from 26 weeks' gestation did not affect our findings.

Results:

1. For ease of understanding and comparison of the two separate cohorts, a more classic table 1 would help. While the information is currently in both the labeled tables 1 and 2, a summary of the overall and vaccinated groups for both would be helpful.

Author response: We hope that the revised text in the first paragraph of the discussion helps to clarify the extent to which these cohorts overlap

"Many women were eligible to be offered both pertussis and influenza vaccinations during the study: 66,143 women were included in both analytic samples (97.1% of the pertussis vaccine cohort and 43.5% of the influenza cohort)."

For this reason, it is perhaps unsurprising that the cohorts have extremely similar baseline characteristics – we looked at including an additional Table 1, but this added repetition to the tables which we felt did not help to address the other key feedback of making the main results clearer. We hope this is acceptable, and would be happy to add a classic table 1 combining the first column of each of tables 1 and 2 as a supplementary table if needed.

2. It would be helpful to have one figure that conveys the major findings more easily to readers.

Author response: Thank you for this helpful suggestion. We have added figure 1 to illustrate the large inequality in vaccine uptake by deprivation for pertussis and influenza, as this is a key finding from the study.

Discussion:

1. For a non-English reader, the significance of these regions is unknown yet is a major focus. If the authors believe that the geographical differences are of significant clinical value, it could be helpful to explain these differences for non-English readers. Could the authors provide somewhere (perhaps the discussion?) why they think the regional variations exist?

Author response: From an English perspective, the regional variation we found is mostly of interest as indicating that our findings align with national surveillance (which reports regional variation), and so we have added a brief comment that vaccine uptake is lower in London across the vaccination programme but have dwelled more on the new information on inequalities which our study adds. We hope that the edited discussion helps to highlight the key points from our study more clearly.

2. The majority of the discussion seems to be further explanation of their data instead of any policy recommendation or next steps. Currently, the discussion only includes vague action items of how to improve in future without clear future directions.

Author response: Thank you – we have edited the discussion with the aim of providing clearer recommendations for policy and research, as described above in response to your global comments.

3. Also, a more in depth review of the differences between deprivation/poverty and race would help the reader understand how independent these variables are.

Author response: In our study, we found that the associations between deprivation and vaccine uptake, and between ethnicity and vaccine uptake appeared to be largely independent of one another. This was surprising, and we highlight it in our discussion, but are cautious of over-interpretation since ethnicity, region and local deprivation are not unrelated in England (in common with many other settings).

Minor concerns:

1. Abstract: Line 42-3, the percentages are in the wrong order compared to the descriptions.

Author response: Thank you for pointing this out, we have remedied this.

2. Abstract: line 52: the word 'substantially' is superfluous and we would recommend deleting.

Author response: We have changed this to 'broadly' to clarify that the associations were not meaningfully different before and after mutual adjustment, without wanting to suggest that the results were exactly the same after adjustment.

3. Article summary: the 4th bullet does not make sense. Please clarify?

Author response: Thank you, we have edited this to read:

"We were unable to investigate vaccine uptake inequalities from 2016 onwards due to the lack of reliable data on vaccination in secondary care settings."

We hope this is clearer.

4. Introduction: truly minor- there is an extra space in line 20.

Author response: thank you, we have now corrected this.

5. Methods: truly minor- there is an extra space on page 10, in line 41.

Author response: Many apologies, despite the very clear description we are unable to find this in the word manuscript. If accepted, we hope that this could be remedied at proofing stage.

6. Results: First line of sample characteristics, line 9, even though it takes more words, I would keep the data for pertussis and influenza together.

Author response: Thank you, we have edited this to read:

"A total of 68,090 women from 402 general practices were eligible for the pertussis vaccine analysis, and 152,132 women from 456 general practices were eligible for the influenza vaccine analysis during the study period."

7. Results: line 16- I would add the % of women in both cohorts.

Author response: Thank you, we have added this so that it now reads:

"Many women were eligible to be offered both pertussis and influenza vaccinations during the study: 66,143 women were included in both analytic samples (97.1% of the pertussis vaccine cohort and 43.5% of the influenza cohort)."

8. Results: Line 33 on page 15 beginning with "Vaccine..." should include a p-value for clarity.

Author response: Thank you, we have started a separate paragraph for the comparison of women with missing ethnicity to women with recorded ethnicity, which now ends with "(all $p < 0.001$, Supplementary Table 2)".

9. Discussion: page 21, line 43, suggest should be singular.

Author response: thank you, we have corrected this.

10. Supplementary Table 5 headings appear to be off center.

Author response: thank you, we hope that the tables remain clearly labelled.

11. In Tables, it would be helpful to clarify that it is 95% CI in parentheses.

Author response: thank you, we have clarified this in both main tables.

12. Also in the tables, bolding the Confidence intervals that do not cross 1.0 would be helpful since there are so many data points to digest. A visual/bar graph of these associations could be helpful.

Author response: Thank you for this helpful suggestion – we have added a bar graph (Figure 1) to draw attention to our main finding, of the large difference in vaccine coverage by deprivation.

VERSION 2 – REVIEW

REVIEWER	Howe, Anna University of Canterbury, Health Sciences
REVIEW RETURNED	18-Apr-2021

GENERAL COMMENTS	Thank you for addressing my concerns.
---------------------------------------

REVIEWER	Thompson, Lindsay University of Florida College of Medicine, Pediatrics
REVIEW RETURNED	26-Apr-2021

GENERAL COMMENTS	Title: Social Determinants of pertussis and influenza vaccine uptake in pregnancy: a national cohort study using electronic health records. Summary: We appreciate the summative work the authors have done to improve this article on what social factors predict vaccination during pregnancy. Its strengths are its completeness of documented analysis as well as in the subsequent analysis of second deliveries- this is a very surprising problem of under-vaccination. Main concerns: Abstract: 1. After reading a few times, it becomes clear why the results start with ‘vaccine uptake in the first eligible pregnancy....’ but it would be more understandable to talk about overall rates since you later talk about parity. Article Summary: 1. Bullet #4 seems like too much of a nuance to highlight here. Introduction: 1. We believe the authors mixed up the dates of the pertussis and influenza vaccination onset dates. (flu= 2010 and pertussis in 2012, according to their citations). Methods: 1. Does the CPRD/LSHTM register include high risk deliveries? Do these stay in primary care? (line 11, page 8) Results: 1. We remain concerned at the large number of supplements but leave it to editorial discretion. 2. Why are the two samples so different in size? (n=68k for pertussis) and n=152k for influenza? (page 16, first sentence.) If this is due to the # available dates, remind the reader here. Discussion: 1. We recommend commenting why the overall influenza rate is so terribly low (39.1%). Minor concerns: 1. Article Summary: Bullet #2 would be better if it started with a subject rather than ‘it.’ Perhaps combine bullets 1 and 2? 2. Introduction: We recommend inserting ‘at no cost’ on line 13 of the first page/paragraph. 3. Methods, page 9, line 11. We would add “women SHOULD be offered...” 4. Methods, page 9, line 54- how many women registered after the first trimester and were excluded?
---

	5. Results, page 18, line 52. We recommend quantifying the lower uptake seen in women of non-white ethnicity for the influenza vaccine. 6. Discussion, page 20, line 46. We would spell out general practitioners since “GPs” is not used elsewhere. 7. Figure 1, page 34. We recommend changing the horizontal axis title to read 1 (least deprived), 2, 3, 4, 5 (most deprived) for clarity.
--	---

VERSION 2 – AUTHOR RESPONSE

Reviewer: 1

Dr. Anna Howe, University of Canterbury, University of Auckland

Comments to the Author:

Thank you for addressing my concerns.

Response: Thank you for your review. We are grateful for your comments, and pleased we were able to address them to improve the manuscript.

Reviewer: 2

Dr. Lindsay Thompson, University of Florida College of Medicine

Comments to the Author:

RE: MS# bmjopen-2020-046545, revision 1.

Summary: We appreciate the summative work the authors have done to improve this article on what social factors predict vaccination during pregnancy. Its strengths are its completeness of documented analysis as well as in the subsequent analysis of second deliveries- this is a very surprising problem of under-vaccination.

Response: Thank you, we are grateful for your comments to help improve the article.

Main concerns:

Abstract:

1. After reading a few times, it becomes clear why the results start with ‘vaccine uptake in the first eligible pregnancy....’ but it would be more understandable to talk about overall rates since you later talk about parity.

Response: We have simplified this to “Vaccine uptake was 67.3% for pertussis, and 39.1% for influenza.” and hope that this is now a clearer summary of the headline study findings.

Article Summary:

1. Bullet #4 seems like too much of a nuance to highlight here.

Response: We have changed this to “We were unable to investigate other potential social determinants not routinely recorded in primary care records such as education and religion.” We hope this presents a more relevant limitation for the article summary.

Introduction:

1. We believe the authors mixed up the dates of the pertussis and influenza vaccination onset dates. (flu= 2010 and pertussis in 2012, according to their citations).

Response: Thank you for spotting this error! We have gratefully corrected it.

Methods:

1. Does the CPRD/LSHTM register include high risk deliveries? Do these stay in primary care? (line 11, page 8)

Response: The register aims to capture all pregnancies known to the general practice, not just deliveries managed in primary care. A previous validation of the CPRD/LSHTM register against Hospital Episode Statistics (HES) data from secondary care found that 90.7% of deliveries recorded in HES had a corresponding CPRD Register delivery record. To clarify the scope of the pregnancy register, we have added to the Data sources section in the methods that:

“The Pregnancy Register has been found to have a high sensitivity for livebirths (including 90% of all deliveries recorded in secondary care) but may under-record pregnancies which end in a loss”

Results:

1. We remain concerned at the large number of supplements but leave it to editorial discretion.

Response: We appreciate we have presented a number of sensitivity analyses to understand how estimates of vaccine uptake and associations with social determinants might be affected by missing data or study inclusion criteria. We feel that providing the results as a supplement allows us to succinctly summarise in the article text that the directions of associations and conclusions were robust to all sensitivity analysis for both vaccines.

2. Why are the two samples so different in size? (n=68k for pertussis) and n=152k for influenza? (page 16, first sentence.) If this is due to the # available dates, remind the reader here.

Response: This is partly because the maternal pertussis vaccination programme started two years later than the maternal influenza vaccine programme, but also because pregnancies ending before 26 weeks gestation would be eligible for influenza vaccination but not pertussis vaccination. We have taken the opportunity to remind the reader of the different study periods:

“A total of 68,090 women from 402 general practices were eligible for the pertussis vaccine analysis, and 152,132 women from 456 general practices were eligible for the influenza vaccine analysis during the study period (2012 to 2015 for pertussis and 2010/11 to 2015/16 for influenza).”

Discussion:

1. We recommend commenting why the overall influenza rate is so terribly low (39.1%).

Response: We have added to the second paragraph of the discussion that:

“Both the overall uptakes and the patterns of regional variation are consistent with national surveillance and ecological studies. Lower vaccine uptake in London is seen more widely across the vaccination programme.^{10, 11, 17, 18} For influenza vaccine, the denominator may be seen as overinclusive as some women may have only a short time period eligible for

vaccination (due to pregnancy loss or limited overlap of pregnancy with influenza season), resulting in a low estimate of uptake.”

Minor concerns:

1. Article Summary: Bullet #2 would be better if it started with a subject rather than ‘it.’ Perhaps combine bullets 1 and 2?

Response: We have combined bullets 1 and 2 as you suggest.

2. Introduction: We recommend inserting ‘at no cost’ on line 13 of the first page/paragraph.

Response: We have clarified that both vaccines are offered to pregnant women at no cost to the woman:

“In England, pertussis vaccination has been offered to women in later stages of pregnancy since 2012 and seasonal influenza vaccination at any stage of pregnancy during influenza season since 2010, with both provided free of charge.”

3. Methods, page 9, line 11. We would add “women SHOULD be offered...”

Response: we have edited the second paragraph within the study population section of the methods to read:

“Vaccination guidelines during the study period suggested women should be offered pertussis vaccination in their third trimester of pregnancy...”

4. Methods, page 9, line 54- how many women registered after the first trimester and were excluded?

Response: Sensitivity analyses in which we included women who were registered after the first trimester are presented in Supplementary Tables 5 (pertussis vaccination) and 6 (influenza vaccination).

- For pertussis vaccination this included 80,831 women (adding 18,294 women to the main analysis, in which women registered after the first trimester were excluded).
- For influenza vaccination this included 153,782 women (adding 13,641 women to the main analysis, in which women registered after the first trimester were excluded).
- For both vaccines, the analysis results were unchanged by including women registered after the first trimester.

5. Results, page 18, line 52. We recommend quantifying the lower uptake seen in women of non-white ethnicity for the influenza vaccine.

Response: We have added to the results summary for influenza vaccine uptake that:

“Influenza vaccination uptake was lowest among women of Black ethnicity, with 16% reduced odds of uptake compared to those of White ethnicity.”

6. Discussion, page 20, line 46. We would spell out general practitioners since “GPs” is not used elsewhere.

Response: We have amended GPs to read general practitioners.

7. Figure 1, page 34. We recommend changing the horizontal axis title to read 1 (least deprived), 2, 3, 4, 5 (most deprived) for clarity.

Response: We have made this change to the axis in figure 1.